# Research

behaviour, cognition, evolution

*Papio*, primate communication, vocal learning, sensory–motor integration, speech evolution, implicit learning

**Author for correspondence:**
Julia Fischer
e-mail: jfischer@dpz.eu

# Vocal convergence in a multi-level primate society: insights into the evolution of vocal learning

Julia Fischer[1,3,4], Franziska Wegdell[1,4], Franziska Trede[1,2], Federica Dal Pesco[1,4] and Kurt Hammerschmidt[1,4]

[1]Cognitive Ethology Laboratory, and [2]Primate Genetics Laboratory, German Primate Center, Kellnerweg 4, 37077 Göttingen, Germany
[3]Department of Primate Cognition, Georg August University Göttingen, Göttingen, Germany
[4]Leibniz ScienceCampus Primate Cognition, Göttingen, Germany

  JF, 0000-0002-5807-0074; FW, 0000-0002-3108-2999; FT, 0000-0003-3690-1006; FDP, 0000-0003-2326-1185; KH, 0000-0002-3430-2993

The extent to which nonhuman primate vocalizations are amenable to modification through experience is relevant for understanding the substrate from which human speech evolved. We examined the vocal behaviour of Guinea baboons, *Papio papio*, ranging in the Niokolo Koba National Park in Senegal. Guinea baboons live in a multi-level society, with units nested within parties nested within gangs. We investigated whether the acoustic structure of grunts of 27 male baboons of two gangs varied with party/gang membership and genetic relatedness. Males in this species are philopatric, resulting in increased male relatedness within gangs and parties. Grunts of males that were members of the same social levels were more similar than those of males in different social levels ($N = 351$ dyads for comparison within and between gangs, and $N = 169$ dyads within and between parties), but the effect sizes were small. Yet, acoustic similarity did not correlate with genetic relatedness, suggesting that higher amounts of social interactions rather than genetic relatedness promote the observed vocal convergence. We consider this convergence a result of sensory–motor integration and suggest this to be an implicit form of vocal learning shared with humans, in contrast to the goal-directed and intentional explicit form of vocal learning unique to human speech acquisition.

## 1. Introduction

One of the key preconditions for the development of speech is the ability to adjust vocal output in response to auditory input. Humans are exceptionally proficient at vocal learning. Although effortless speech learning is confined to the early years [1], humans still possess the ability to imitate sounds voluntarily and acquire further languages throughout their lives. Numerous comparative studies have aimed at elucidating the evolutionary origins of vocal learning within the primate lineage, to uncover the extent to which nonhuman primates reveal evidence for vocal plasticity, and whether such plasticity may be conceived as a pre-adaptation for the evolution of speech [2,3].

Despite considerable research effort, it appears that the ability to learn sounds from auditory experience in most nonhuman primate species is limited. Unlike humans or some songbird species, nonhuman primates are not obligatory vocal learners that require species-specific auditory input to develop their normal vocal repertoires [4,5]. Early attempts to train a young chimpanzee to produce speech sounds yielded disappointing results and prompted most of the 'ape language' projects to turn to another modality, using either symbol systems or sign languages [6]. Studies of the neural basis of vocal production in different monkey species found that the animals lack the neural connections necessary

**Figure 1.** (a) The multi-level social organization of Guinea baboons. Several units form a party, and two or more parties form a gang. (b) Spectrogram of grunts from four different males. Frequency (kHz) on the y-axis, time (s) on the x-axis. The spectrogram was created using Avisoft-SASLab Pro 5.2 (1.024 pt FFT, sampling frequency: 11 kHz, time resolution: 2.9 ms, Flat Top window). (Online version in colour.)

for the volitional control over the fine structure of vocalizations, although they exert greater control over the usage of calls (reviewed in [2]). One exception to the rule of limited vocal plasticity may be orangutans, which have greater control over their vocal apparatus [7,8].

In addition to the limited ontogenetic plasticity, a range of comparative studies within different nonhuman primate species strongly suggest that the motor patterns underlying vocalizations are evolutionarily highly conserved within genera (reviewed in [2]). For instance, the structure of alarm calls of members of the genus *Chlorocebus* differs only marginally between East African vervets, *Chlorocebus pygerythrus*, and West African green monkeys, *Chlorocebus sabaeus.* Moreover, in response to a drone, naive West African green monkeys spontaneously uttered calls that structurally were highly similar to East African vervet 'eagle alarm calls', indicating that the link between the perception of a specific (potentially dangerous) stimulus and the activation of a given motor programme is also conserved [9].

At the same time, subtle modifications in vocal output as a result of auditory experience appear to be possible. For instance, common marmosets, *Callithrix jacchus,* increase the amplitude of their calls in noisy environments ('Lombard effect'; [10,11]). More importantly, a range of species show group-specific variations or 'dialects' in their vocalizations (reviewed in [12]), while Japanese macaques matched some of the acoustic features of calls presented in playbacks [13]. These instances of vocal plasticity have been described as 'vocal accommodation' [12,14,15] or 'social shaping' [15], similar to the observation that humans may involuntarily match the pitch, temporal patterning and prosody of the people they are talking to.

Following the idea that auditory input may lead to vocal convergence, subjects that interact more frequently with one another using vocalizations should produce calls that are more similar to each other than those that interact less frequently. A higher acoustic similarity may also result from genetic relatedness, however. For instance, highly related subjects may also have a similar morphology of the vocal production apparatus [16]. Before conclusions about the role of experience can be drawn, it is necessary to assess whether potential acoustic variation between individuals can (also) be explained by genetic distance.

To date, few studies have investigated the effects of genetic relatedness and interaction frequency at the same time. Lemasson and colleagues reported that interaction frequency

but not genetic relatedness accounted for acoustic variation in the calls of Campbell's monkeys, *Cercopithecus campbelli campbelli* [17]. The reported correlation of acoustic similarity with grooming frequency may be spurious, however, as data from two groups (with $N = 6$ and 4 females, respectively) were pooled, and the correlation was largely driven by the differences between the groups. Levréro and colleagues [15] studied the acoustic structure of contact calls in 36 male and female mandrills living in three social groups. Both genetic relatedness and familiarity impacted acoustic similarity of the species' 'kakak' calls, while retaining cues to kin memberships: playback experiments showed that subjects responded significantly more strongly to calls recorded from related kin, irrespective of familiarity [15].

We here set out to assess the impact of social interaction while controlling for genetic relatedness by comparing the acoustic variation in the grunts of male Guinea baboons, *Papio papio*. Guinea baboons are an interesting model to examine the influence of auditory experience and social group membership, as they live in a nested multi-level society with male philopatry [18]. At the base of the society are 'units' comprising one adult male, one to six females and young. A small number of units, together with bachelor males, form a 'party', and two or three parties make up a 'gang' (figure 1a). Assignments to parties and gangs are based on spatial proximity and affiliative interactions [19]. During affiliative interactions with other group members, males produce low-frequency tonal grunts (figure 1b). The Guinea baboons' social structure allowed us to assess vocal convergence at two social levels, namely within parties and within gangs.

If the frequency of interaction affects the structure of calls, subjects that interact frequently with one another should produce calls that are more similar to each other. Thus, members of the same party should have the greatest similarity, while members of the same gang should produce calls that are more similar to each other than to calls produced by members of another gang. If genetic relatedness affects the vocal structure, dyads that are more highly related should reveal greater acoustic similarity. Note that these two effects (interaction frequency and relatedness) are not mutually exclusive.

## 2. Methods

We obtained recordings of grunts from a total of 27 male baboons in 2010/11, 2014 and 2016. Thirteen of the males were members of the 'Mare' gang and 14 were members of the 'Simenti' gang. The

Mare gang comprised two parties of 6 and 7 males each; the Simenti gang comprised two parties of 5 and 9 males each. Twenty-three of the 27 males were confirmed or assumed to be present throughout the study period (see electronic supplementary material, table S1).

Vocalizations were recorded using Marantz PMD 661 recorders (D&M Professional, Longford, UK) with Sennheiser directional microphones (K6 power module + ME66 recording head; Sennheiser, Wedemark, Germany) equipped with Rycote windshields (Rycote Windjammer, Rycote Ltd, Stroud, UK). We used Avisoft-SASLab Pro (Avisoft Bioacoustic, Berlin, Germany) to check the recording quality and to label and extract grunts with sufficient quality and low background noise. We only used calls recorded at a maximum distance of 3 m. To maximize the number of grunts per male, we included grunts from different contexts (electronic supplementary material, table S2). In total, we included 756 grunts in the acoustic analysis. On average, we used 28 calls per subject in the analysis (range: 5–127). The Mare and Simenti gang males were represented by 390 and 366 grunts, respectively. Ideally, one would have liked to include further gangs with additional subjects to assess whether the observed pattern holds beyond our study population, but adding further groups was beyond our capacities.

We reduced the sampling frequency from 44.1 to 5 and 1 kHz to obtain an appropriate frequency resolution for the estimation of acoustic features and calculated two 1024-pt fast Fourier transformations (FFTs), one resulting in a frequency range of 2500 Hz (frequency resolution 5 Hz, temporal resolution 6.4 ms) and a second FFT resulting in a frequency range of 500 Hz (frequency resolution of 1 Hz, and a temporal resolution of 16 ms). Calculating two FFTs allowed us to maximize the temporal resolution for the entire call type, and estimate the fundamental frequency at a higher frequency resolution. The resulting frequency–time spectra were analysed with a custom software program LMA 2019, which allows visual control of the accuracy of parameter estimation [20,21]. LMA outputs a total of 82 acoustic parameters.

To identify which parameters would be informative to distinguish between individuals (and thus, social levels), we entered all 82 acoustic features from the LMA output into a stepwise discriminant function (DFA) with subject identity as a grouping variable. The selection criterion for acoustic features to enter the discriminant function analysis was $P_{in} = 0.05$ and to be removed $P_{out} = 0.1$. The DFA used 31 acoustic features for individual discrimination (electronic supplementary material, table S3). To quantify the acoustic distance between males, we used the average pairwise $F$-value from the discriminant function analysis as a dissimilarity score for each dyad. The dissimilarity score provides an assessment of the similarity of calls, with higher values indicating greater dissimilarity and lower values greater similarity. In the following, we will simply refer to the similarity of calls. The average pairwise $F$-value has been used in different studies examining relationships between acoustic structure and genetic or geographic distance [22,23]. The discriminant function analysis was performed using IBM SPSS v. 26.0 (IBM, Armonk, NY). To assess whether the classification result of the individual discrimination of male grunts is higher than would be expected by chance, we additionally performed a permuted DFA [24], which controls for variation in individual contributions of grunts.

We extracted DNA from faecal samples using the First-DNA all tissue kit (Genial®) and characterized genetic variation by assessing the individual allele variation on 24 polymorphic autosomal microsatellite markers. The 24 markers were amplified using the Multiplex PCR Kit (QIAGEN) and fluorescent-labelled primers. PCR products were separated and detected through capillary gel electrophoresis on an ABI 3130xL Genetic Analyzer (Applied Biosystems®, USA). Microsatellite allele sizes were evaluated using GeneMapper 5 (Applied Biosystems®). One locus (D1s548) showed signs of null alleles and significant deviations

from Hardy–Weinberg equilibrium and was therefore excluded, resulting in a total of 23 loci included in the relatedness estimation (calculated with MICRO-CHECKER v. 2.2.3 [25] and the PopGenReport R package v. 3.0.0 [26]. We used the R package 'related' v. 1.0 [27,28] to estimate relatedness using R v. 3.4.4 and RStudio v. 1.1.456. The Wang estimator (hereafter $W$) appeared to be most suitable for the present analysis (see electronic supplementary material, table S4). $W$ ranges from −1 to 1. Negative values indicate that dyads are less related than on average, while positive values indicate that they are more highly related than on average (see [29] for a detailed description of the analysis).

These and the following statistical analyses were conducted in the R environment v. 3.6.3 [30], using the RStudio interface v. 1.3.959 [31]. We used a Mantel matrix correlation test (package 'vegan'; v. 2.5.6) to test the correlation between acoustic and genetic variation. To test whether calls within a gang were more similar to each other than between gangs, we applied a categorical Mantel test, using 'same gang membership' (yes/no) as the categorical predictor variable, and $W$ or $F$ (transformed as $\ln(1 + F)$) as the continuous variable. The analysis of the effect of gang membership was based on 351 dyads. To study the effect of party membership, we also used a categorical Mantel test, but only considered pairs of males that lived in the same gang (e.g. SNE and BAA, both members of the Mare gang, or BEN and WLD, both members of the Simenti gang; total number of dyads within both gangs, $N = 169$). We used a restricted permutation approach where males were permuted between parties within gangs. We used 1000 permutations in all analyses, except the one for the variation between parties within gangs, where we used 10 000 permutations. Effect sizes were calculated with the package 'effsize' version 0.8.0. The data and code for statistical analysis are deposited at https://osf.io/h7q5r/.

## 3. Results

Confirming previous analyses, males were more highly related within gangs than between gangs (categorical Mantel test, $p = 0.001$, $N = 351$; figure 2a). The effect size (Cohen's $d$) was 0.52 (CI$_{lower}$ −0.73, CI$_{upper}$, −0.31; medium effect size). Within gangs, males in the same party were more highly related on average than males that were not members of the same party ($p = 0.035$, $N = 169$; figure 2b), with a small effect size ($d = 0.24$, CI$_{lower}$ −0.54, CI$_{upper}$ −0.07).

Grunts could be assigned to the correct individual significantly more frequently than by chance, with an average correct assignment of 34.5% using the procedure in SPSS (chance level 3.7%, leave-one-out validation: 21.0% correct classification). The classification in the permuted DFA (pDFA) with a reduced set of $N = 135$ calls (see electronic supplementary material, table S5) yielded an average classification of 11.2% ($p < 0.001$). Acoustic similarity did not correlate with genetic similarity ($r = −0.006$, $p = 0.515$). Because of the inherent uncertainty with which dyadic relatedness can be estimated [32], we ran an additional analysis in which we compared the acoustic similarity of dyads in the top quartile ($W > 0.125$) versus the bottom quartile ($W < −0.117$). Again, we found no effect of relatedness on acoustic similarity (categorical Mantel test, $p = 0.933$; figure 3).

Grunts of males within gangs were more similar to each other than between gangs (categorical Mantel test, $p = 0.012$; figure 4a), and grunts of males within a party were also more similar to each other than between parties in the same gang ($p = 0.001$; figure 4b). The effect sizes were modest,

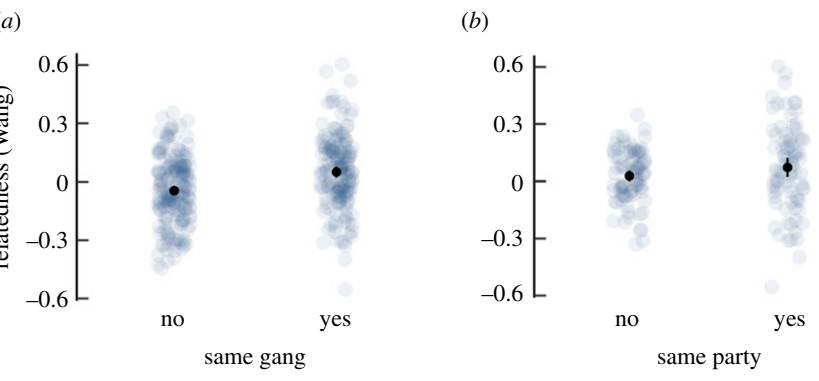

**Figure 2.** Genetic relatedness between male dyads that belong to (*a*) different gangs or the same gang and (*b*) different parties or the same party within a gang. Light grey dots represent dyadic values, black dots the mean with 95% confidence interval. (Online version in colour.)

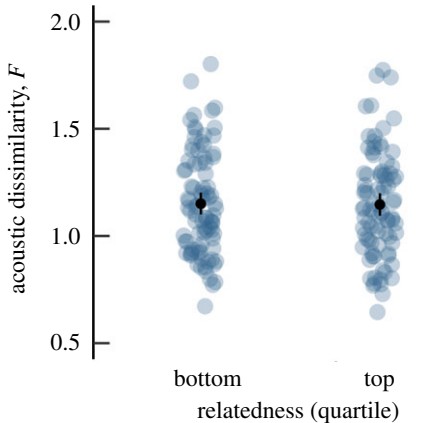

**Figure 3.** Relation between acoustic dissimilarity and genetic relatedness (top and bottom quartiles of the Wang estimator $W$) for $N = 175$ dyads. Note that lower dissimilarity values indicate higher similarity. Light grey dots represent dyadic values, black dots the mean with 95% confidence interval. (Online version in colour.)

however ($d = 0.177$, $CI_{lower}$ −0.03, $CI_{upper}$ 0.38 between gangs and 0.152, $CI_{lower}$ −0.15, $CI_{upper}$ 0.46 between parties, respectively). When we compared the mean acoustic similarity of males that resided in the same party (mean $\log F = 0.29$) with those that were part of a different gang (mean $\log F = 0.33$), the effect size was small ($d = 0.24$, $CI_{lower}$ −0.02, $CI_{upper}$ 0.50). Grunts varied with social level (party/gang) mostly in parameters that are related to the filter function of the vocal tract (electronic supplementary material, table S6). The fundamental frequency or call duration did not vary systematically between social levels.

## 4. Discussion

The structure of male grunts varied between members of different gangs, and also between members of parties within a gang. The effect sizes of these two comparisons were modest, however. Males in the same gang were also more highly related to one another, but this did not account for the acoustic variation between parties and gangs, as evidenced by the lack of an effect of genetic relatedness on acoustic similarity. In this regard, the Guinea baboons differ from mandrills, where both relatedness and interaction frequency predicted the structure of the vocalizations [15].

It may seem puzzling at first that genetic relatedness did not account for the higher vocal similarity in Guinea baboons despite the fact that genetic relatedness and acoustic similarity were both higher within parties and gangs than between. This can be explained by the fact that not all dyads within a social level are indeed more highly related than across these social levels. Acoustic similarity thus appears mainly to be driven by social interaction, which is not restricted to highly related dyads. To a certain degree, relatedness and acoustic similarity vary independently of one another.

How may auditory input affect vocal production? One mechanism that may support the reported minor adjustments in vocal output with experience may rest on sensory–motor integration [33]. According to the idea of a 'common coding' framework, specific sensorimotor areas represent both sensory input and motor commands generating that corresponding pattern [34]. In humans, neuroimaging studies identified specific motor activations when subjects listened to speech sounds [35]. If such sensory–motor integration exists in the auditory–vocal domain of nonhuman primates, the exposure to specific auditory input may increase the likelihood to produce the corresponding motor pattern via co-activation.

A recent study provided compelling evidence for the integration of auditory input with vocal output in a nonhuman primate species. In common marmosets, activity in the auditory cortex directly affected the monkeys' control of vocal production [36]. Firstly, a shift in the auditory feedback of the monkeys' vocalization led to compensatory changes in the frequency patterns of the subsequent vocalizations. Secondly, microstimulation of the auditory cortex during vocalization led to abrupt shifts in vocal frequency [36]. In a translocation experiment, common marmosets ($N = 4$) adjusted the structure of their vocalizations in response to auditory input from conspecifics, even if the individuals did not interact directly [37]. Beyond the immediate effects of auditory experience, there is also evidence that feedback from parents affects the trajectory of vocal development in marmosets [38–40].

It has been argued that the human ability to imitate the utterances of others gradually evolved from the vocal plasticity observed in nonhuman primates [17,41]. We contend that vocal learning may be based on a variety of different mechanisms, including vocal convergence, 'learning from success', a form of usage learning that comprises the use of specific call variants because they are more likely to yield the desired response, as well as the spontaneous imitation of a recently formed auditory template [42]. Instead of conceiving vocal learning capacities as a continuum [43], we agree with other authors that vocal learning may be supported by a variety of different mechanisms [44]. Future

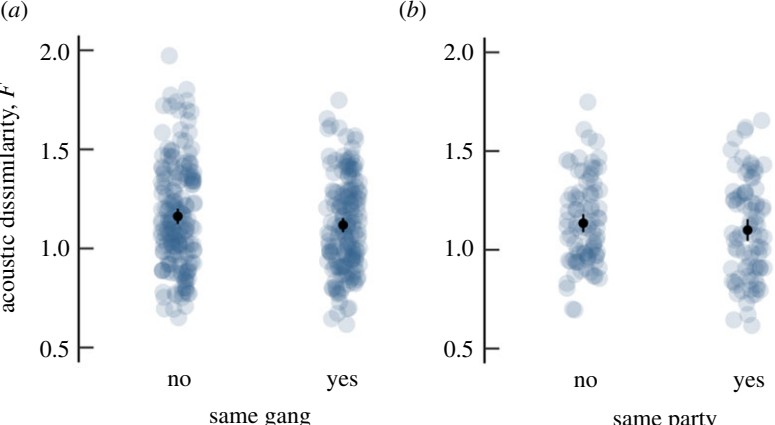

**Figure 4.** Acoustic dissimilarity of dyads that belong to (*a*) different gangs or the same gang and (*b*) different parties or the same party within a gang. Lower dissimilarity values indicate higher acoustic similarity. Light grey dots represent dyadic values, black dots the mean with 95% confidence interval. Calls from males in the same gang and the same party were on average more similar to each other than between gangs or parties. (Online version in colour.)

studies should aim to distinguish between these mechanisms, and also consider the effect size of vocal plasticity.

Taking the extent of plasticity as well as the mechanisms that support them into account will contribute to overcoming futile debates about whether or not nonhuman primates reveal evidence for vocal learning [45]. The vast majority of studies in nonhuman primates that reported evidence for vocal convergence observed only minor changes within the species-specific range of calls. Thus, the small effect sizes reported here are important aspects of the results. Humans, instead, are not only able to work on their accents, as Eliza Doolittle in '*My fair lady*', but they can also sing 'supercalifragilisticexpialidocious' with Mary Poppins. The spontaneous imitation of new words is open-ended, while it is much more difficult to change one's accent once a certain age has been reached. Vocal convergence in nonhuman primates appears to be more similar to the formation of an accent than the acquisition of novel phonetic combinations that make up new words. An interesting open question is whether vocal convergence is simply a by-product of the sensory experience or whether it has been selected for, since it may signal 'in-group' membership and thus have an important social function [46].

Irrespective of whether vocal convergence has been selected for or not, we propose that it constitutes an implicit form of motor learning shared between nonhuman primates and humans, while speech production constitutes an explicit form of motor learning. Implicit and explicit processes are not entirely dichotomous: explicitly acquired motor skills can become automatic (as when you learn to drive a car), while implicit processes may be made explicit [47]. Yet, it has proven useful to distinguish between implicit and explicit forms of knowledge and knowledge acquisition [48]. Taatgen suggested that implicit learning is a by-product of general learning mechanisms, while explicit learning is tied to learning goals and thus intentionality [47]. This definition appears useful for the distinction between vocal convergence as a result of sensory–motor integration on the one hand and the goal-directed acquisition of the patterns that result in the production of speech, on the other.

A further open question is whether the observed acoustic variation is salient to the animals themselves. In a previous study [49], we tested male responses to the playbacks of grunts of males that share the same home range as the study males (neighbours) versus to grunts of males living 50 km away

(strangers). As a control, we played back the grunts of males from their own gang. Surprisingly, males responded strongly only to the grunts from males of their own gang, but largely ignored neighbour or stranger males' calls. In principle, these responses could be explained by the recognition of the males' voice characteristics. Yet, it might also be the case that males recognize the 'sound' of their subgroup. Playbacks presenting artificially created grunts bearing the own gang's characteristics versus another gang's characteristics would be needed to test this conjecture.

In summary, we find evidence for a moderate degree of vocal convergence in the grunts of male Guinea baboons. The magnitude of the change is difficult to compare with those of other studies on nonhuman primates mentioned above, given the differences in methodological approaches, but broadly appears to be in a similar range. Our findings add to the body of evidence that within species-specific constraints, subtle and potentially meaningful variation can be found in nonhuman primate vocalizations. This variation does not compare with the open-ended possibility of vocal imitation found in human speech, however.

**Ethics.** This research complies with the Association for the Study of Animal Behaviour Guidelines for the Use of Animals in Research (Animal Behaviour, 2018, 135, I–X), the legal requirements of the country in which the work was carried out, and all institutional guidelines. The research has been approved by the Diréction des Parcs Nationaux of the Republic of Senegal (22 April 2019).

**Data accessibility.** The data and code for statistical analysis are deposited at https://osf.io/h7q5r/.

**Authors' contributions.** J.F., K.H. and F.W. conceived the study, F.W. collected the data, F.W. and K.H. conducted the acoustic analysis, F.D.P. and F.T. conducted the genetic analysis, J.F. and K.H. ran the statistical analyses and prepared the figures, J.F. wrote the manuscript with input from all authors.

**Competing interests.** We declare we have no competing interests.

**Funding.** This research was supported by the Deutsche Forschungsgemeinschaft (DFG, German Research Foundation) – Project no. 254142454/GRK 2070, the Leibniz ScienceCampus Primate Cognition of the Leibniz Association, and the Christian-Vogel Fonds of the Gesellschaft für Primatologie.

**Acknowledgements.** We thank the Diréction des Parcs Nationaux (DPN) and Ministère de l'Environnement et de la Protéction de la Nature (MEPN) de la République du Sénégal for permission to work in the Niokolo Koba National Park. Roger Mundry provided invaluable statistical advice. We are grateful to two anonymous reviewers and the editors for very helpful comments on the manuscript.

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
