## [Reviewer comments · Proceedings of the Royal Society B: Biological Sciences]

Review History

RSPB-2020-1666.R0 (Original submission)

Review form: Reviewer 1

Recommendation

Accept with minor revision (please list in comments)

Scientific importance: Is the manuscript an original and important contribution to its field?

Good

General interest: Is the paper of sufficient general interest?

Good

Quality of the paper: Is the overall quality of the paper suitable?

Good

Is the length of the paper justified?

Yes

Should the paper be seen by a specialist statistical reviewer?

No

Do you have any concerns about statistical analyses in this paper? If so, please specify them explicitly in your report.

No

It is a condition of publication that authors make their supporting data, code and materials available - either as supplementary material or hosted in an external repository. Please rate, if applicable, the supporting data on the following criteria.

Is it accessible?

Yes

Is it clear?

Yes

Is it adequate?

No

Do you have any ethical concerns with this paper?

No

Comments to the Author

This is a beautiful piece of work examining group differences in the acoustic properties of grunts in a population of guinea baboons in Senegal. The authors found that acoustic and genetic similarity did not correlate. The authors provide a clear, interesting discussion drawing from neurobiology to propose potential mechanisms to account for the findings, as well as a helpful discussion contrasting implicit and explicit vocal learning patterns.

Not addressed, but of interest, given that some implicit vocal convergence of individuals within but not between parties seems to occur, is what acoustic features seem to be more amenable to this type of plasticity than others? This information is presumably evident in the acoustic analysis conducted. It would be a useful addition to present this.

Also not addressed, but would be useful for future research in this area, is even a short discussion regarding the purported function of this behavior – what do male guinea baboons gain from a mechanism of vocal convergence with those they engage in more social interactions? In lines 281-282, the authors suggest this may simply be a “by-product of the sensory experience and does not necessarily be selected for”. However, it seems equally parsimonious to suggest that this could have been selected for, especially as hypotheses exist for vocal convergence in other species. Whilst there may not be space to go into this in detail here, it would be important to briefly acknowledge this possibility - unless the authors feel they have a case to discount this possibility. If the latter, it would still be useful to state the author’s line of argument here.

In terms of data access, I could not find any genetic relatedness data, but perhaps I have missed it?

Other minor comments:

L. 113. Perhaps here it would be good to add a sentence to clarify the basis for assigning units to parties and parties to gangs. Presumably this is based on association, rate of social interactions and some social network measure? It would be good clarify this. Is this essentially categorical assignment – ie some units never associate so are not assigned to the same party, or is it more a classification by degree?

L. 193. If I understand correctly, the comparison of grunt similarity within and between parties is conducted only at the gang level and not between gangs? Is there an ecological basis for this eg that baboons from different gangs do not overlap spatially nor acoustically? If there is overlap

between gangs, would it not strengthen the understanding of the results to compare between parties of different gangs? By this I mean that a) if vocal differences between parties arise due to frequent social interactions, then this result should be strengthened by comparing across parties of different gangs. B) if vocal differences between parties drop when including parties from different gangs, this is also interesting.

L. 254. Given that the effect size is small perhaps 'higher' acoustic similarities between parties, is more accurate than 'high'.

L. 282. Word missing in this sentence?

L. 334. In the conclusion, you say with what your results in the guinea baboons do not compare with in humans. I suggest it would be helpful to add here - as it would be in the abstract - what it does compare with in humans ie implicit vocal accommodation, as seen when acquiring speech accents/dialects.

Review form: Reviewer 2

Recommendation

Reject - article is not of sufficient interest (we will consider a transfer to another journal)

Scientific importance: Is the manuscript an original and important contribution to its field?

Good

General interest: Is the paper of sufficient general interest?

Acceptable

Quality of the paper: Is the overall quality of the paper suitable?

Good

Is the length of the paper justified?

No

Should the paper be seen by a specialist statistical reviewer?

Yes

Do you have any concerns about statistical analyses in this paper? If so, please specify them explicitly in your report.

Yes

It is a condition of publication that authors make their supporting data, code and materials available - either as supplementary material or hosted in an external repository. Please rate, if applicable, the supporting data on the following criteria.

Is it accessible?

Yes

Is it clear?

Yes

Is it adequate?

Yes

Do you have any ethical concerns with this paper?

No

Comments to the Author

The authors studied whether vocal similarity in Guinea baboon males is more likely explained by social unit or by genetic relatedness. This is especially interesting as a higher vocal similarity within social unit, independent of relatedness, hints at some vocal flexibility often denied to exist in primates. What the authors found was that indeed vocal similarity was higher within units than between units, and that this seemed to be the case independent of the relatedness between individuals, even though relatedness was also higher within social unit than between.

In general, I liked research question as well as the approach of the study, but I think there are quite a few things that could be clarified both in the manuscript as well as in the analysis to help understanding it even better, see comments below.

Abstract, Line 33: Does social level here refer to the social organization? Spontaneously, as not being a baboon expert, I would have assumed it to be hierarchical levels.

Line 80: common marmosets also show evidence for adjusting to social environment (see Zürcher, Willems & Burkart, 2019, Plos One)

Line 115ff: The affiliative interactions, during which males produce grunts, are they usually directed towards other males, or towards female and offspring? And do females also grunt? I think this is highly important to know. If males mainly grunt towards females, the grunts would not be social interactions directed towards the males, which I think would be the ones we would expect to have an influence on the male's vocalization.

Line 135: Were all males present during the whole study? Were there no changes within the unit or party-composition by males becoming adults or dying? If it was, how was this taken into account in the analysis?

Line 146ff: What was the reason to perform two FFT's? How were the results of the two analyses treated afterwards? Were they somehow merged or both included in the following analyses?

Line 156: "...all of which were uttered in affiliative contexts". Do you have any further information of the context in which they were uttered? Were they all mixed? Again, I think the communication partner here could be quite important.

Line 159ff: When quantifying the acoustic similarity, did you use a call average for each male, or how did you take the non-independence of the calls into account when calculating the F-values?

Line 162: In the supplementary table, I count around 30 parameters. How do you get to 82 in this analysis? And were the parameters transformed in any way to take into account different measurement units?

Line 188: For me, it was somewhat hard to understand on which levels you test the correlation between acoustic and genetic variation. Probably this could be re-formulated a bit to help the readers understand the process better?

Line 194: If I understand this correctly, you only compare the more closely related individuals with each other, by only including the dyads within the same gang. Wouldn't comparing between less closely related individuals actually be more interesting and more meaningful, as there would be more variation in relatedness within which differences could be found? I would expect that, if you find a correlation between relatedness and vocal similarity, it would be between the less related parties of different gangs.

Methods in general:

What about the units? As they are the structure where animals interact the most, it would be interesting to know why you did not include them into your analyses?

Line 223: When checking the "baboons 27males pdfa.txt" file, it shows a different value for percent correctly cross classified (approx.11.21%). Probably worth checking again.

Line 230: This is the first time acoustic dissimilarity comes up. Could you quickly explain what you mean by it and how you calculated it?

Line 231: there is something missing after SD=, as well as the closing bracket

Line 250: Your results are indeed somewhat confusing. Could you elaborate on some ideas here on why this could be the case? I think it would really be worth it to help the reader along, as this

is your main result of the study, but it is really not easy to gasp.

Line 267: See also Gultekin & Hage 2017 on how auditory feedback can be important for proper vocal development.

Line 257 – 316: Even though highly interesting, I find this part in the discussion a bit too long, especially as it is not really linked to the study or the results themselves. Probably this could be condensed a bit in favour of providing potential explanations for the results?

Decision letter (RSPB-2020-1666.R0)

22-Sep-2020

Dear Professor Fischer:

I am writing to inform you that your manuscript RSPB-2020-1666 entitled "Vocal convergence in a multi-level primate society: insights into the evolution of vocal learning" has, in its current form, been rejected for publication in Proceedings B. This action has been taken on the advice of referees, who have recommended that substantial revisions are necessary. However, both of the reviewers, as well as the Associate Editor and myself, are also enthusiastic about the goals of your manuscript and think that with an appropriate revision you may be able to address the issues raised (outlined below, in the comments from the AE and the reviewers). In particular, while any study of this sort will have both a small sample and a modest effect, there needs to be more discussion of how this impacts interpretation of the data. With this in mind we would be happy to consider a resubmission, provided the comments of the referees are fully addressed. However please note that this is not a provisional acceptance.

Sincerely,

Dr Sarah Brosnan
Editor, Proceedings B
mailto: proceedingsb@royalsociety.org

Associate Editor

Comments to Author:

The paper presents findings suggesting that social learning of vocalisations takes place in primate societies. They report similarity of vocalisations that can't be attributed to genetics. This is interesting and has important implications for the evolution of language. However, the effect is small and (although bigger than many other studies) the sample size is still limited. In my view the authors need to comment on this and provide justification. R2 is also concerned about pseudoreplication. My reading is that the DFA takes individual into account so this shouldn't be a major concern, but the authors need to clarify this.

Reviewer(s)' Comments to Author:

Referee: 1

Comments to the Author(s)

This is a beautiful piece of work examining group differences in the acoustic properties of grunts in a population of guinea baboons in Senegal. The authors found that acoustic and genetic similarity did not correlate. The authors provide a clear, interesting discussion drawing from neurobiology to propose potential mechanisms to account for the findings, as well as a helpful discussion contrasting implicit and explicit vocal learning patterns.

Not addressed, but of interest, given that some implicit vocal convergence of individuals within but not between parties seems to occur, is what acoustic features seem to be more amenable to this type of plasticity than others? This information is presumably evident in the acoustic analysis conducted. It would be a useful addition to present this.

Also not addressed, but would be useful for future research in this area, is even a short discussion regarding the purported function of this behavior – what do male guinea baboons gain from a mechanism of vocal convergence with those they engage in more social interactions? In lines 281-282, the authors suggest this may simply be a “by-product of the sensory experience and does not necessarily be selected for”. However, it seems equally parsimonious to suggest that this could have been selected for, especially as hypotheses exist for vocal convergence in other species. Whilst there may not be space to go into this in detail here, it would be important to briefly acknowledge this possibility - unless the authors feel they have a case to discount this possibility. If the latter, it would still be useful to state the author's line of argument here.

In terms of data access, I could not find any genetic relatedness data, but perhaps I have missed it?

Other minor comments:

L. 113. Perhaps here it would be good to add a sentence to clarify the basis for assigning units to parties and parties to gangs. Presumably this is based on association, rate of social interactions and some social network measure? It would be good clarify this. Is this essentially categorical assignment – ie some units never associate so are not assigned to the same party, or is it more a classification by degree?

L. 193. If I understand correctly, the comparison of grunt similarity within and between parties is conducted only at the gang level and not between gangs? Is there an ecological basis for this eg that baboons from different gangs do not overlap spatially nor acoustically? If there is overlap between gangs, would it not strengthen the understanding of the results to compare between parties of different gangs? By this I mean that a) if vocal differences between parties arise due to frequent social interactions, then this result should be strengthened by comparing across parties of different gangs. B) if vocal differences between parties drop when including parties from different gangs, this is also interesting.

L. 254. Given that the effect size is small perhaps 'higher' acoustic similarities between parties, is more accurate than 'high'.

L. 282. Word missing in this sentence?

L. 334. In the conclusion, you say with what your results in the guinea baboons do not compare with in humans. I suggest it would be helpful to add here – as it would be in the abstract - what it does compare with in humans ie implicit vocal accommodation, as seen when acquiring speech accents/dialects.

Referee: 2

Comments to the Author(s)

The authors studied whether vocal similarity in Guinea baboon males is more likely explained by social unit or by genetic relatedness. This is especially interesting as a higher vocal similarity within social unit, independent of relatedness, hints at some vocal flexibility often denied to exist in primates. What the authors found was that indeed vocal similarity was higher within units than between units, and that this seemed to be the case independent of the relatedness between individuals, even though relatedness was also higher within social unit than between.

In general, I liked research question as well as the approach of the study, but I think there are quite a few things that could be clarified both in the manuscript as well as in the analysis to help understanding it even better, see comments below.

Abstract, Line 33: Does social level here refer to the social organization? Spontaneously, as not being a baboon expert, I would have assumed it to be hierarchical levels.

Line 80: common marmosets also show evidence for adjusting to social environment (see Zürcher, Willems & Burkart, 2019, Plos One)

Line 115ff: The affiliative interactions, during which males produce grunts, are they usually directed towards other males, or towards female and offspring? And do females also grunt? I think this is highly important to know. If males mainly grunt towards females, the grunts would not be social interactions directed towards the males, which I think would be the ones we would expect to have an influence on the male's vocalization.

Line 135: Were all males present during the whole study? Were there no changes within the unit or party-composition by males becoming adults or dying? If it was, how was this taken into account in the analysis?

Line 146ff: What was the reason to perform two FFT's? How where the results of the two analyses treated afterwards? Where they somehow merged or both included in the following analyses?

Line 156: "...all of which were uttered in affiliative contexts". Do you have any further information of the context in which they were uttered? Were they all mixed? Again, I think the communication partner here could be quite important.

Line 159ff: When quantifying the acoustic similarity, did you use a call average for each male, or how did you take the non-independence of the calls into account when calculating the F-values?

Line 162: In the supplementary table, I count around 30 parameters. How do you get to 82 in this analysis? And were the parameters transformed in any way to take into account different measurement units?

Line 188: For me, it was somewhat hard to understand on which levels you test the correlation between acoustic and genetic variation. Probably this could be re-formulated a bit to help the readers understand the process better?

Line 194: If I understand this correctly, you only compare the more closely related individuals with each other, by only including the dyads within the same gang. Wouldn't comparing between less closely related individuals actually be more interesting and more meaningful, as there would be more variation in relatedness within which differences could be found? I would expect that, if you find a correlation between relatedness and vocal similarity, it would be between the less related parties of different gangs.

Methods in general:

What about the units? As they are the structure where animals interact the most, it would be interesting to know why you did not include them into your analyses?

Line 223: When checking the “baboons 27males pdfa.txt” file, it shows a different value for percent correctly cross classified (approx.11.21%). Probably worth checking again.

Line 230: This is the first time acoustic dissimilarity comes up. Could you quickly explain what you mean by it and how you calculated it?

Line 231: there is something missing after SD=, as well as the closing bracket

Line 250: Your results are indeed somewhat confusing. Could you elaborate on some ideas here on why this could be the case? I think it would really be worth it to help the reader along, as this is your main result of the study, but it is really not easy to gasp.

Line 267: See also Gultekin & Hage 2017 on how auditory feedback can be important for proper vocal development.

Line 257 – 316: Even though highly interesting, I find this part in the discussion a bit too long, especially as it is not really linked to the study or the results themselves. Probably this could be condensed a bit in favour of providing potential explanations for the results?

Author's Response to Decision Letter for (RSPB-2020-1666.R0)

See Appendix A.

RSPB-2020-2531.R0

Review form: Reviewer 1

Recommendation

Accept as is

Scientific importance: Is the manuscript an original and important contribution to its field?

Good

General interest: Is the paper of sufficient general interest?

Good

Quality of the paper: Is the overall quality of the paper suitable?

Excellent

Is the length of the paper justified?

Yes

Should the paper be seen by a specialist statistical reviewer?

No

Do you have any concerns about statistical analyses in this paper? If so, please specify them explicitly in your report.

No

It is a condition of publication that authors make their supporting data, code and materials available - either as supplementary material or hosted in an external repository. Please rate, if applicable, the supporting data on the following criteria.

Is it accessible?

Yes

Is it clear?

Yes

Is it adequate?

Yes

Do you have any ethical concerns with this paper?

No

Comments to the Author

I appreciated the changes the authors have made. I find the revisions clear, appropriate and well done.

I would suggest only one small change. It is very helpful to have the additional information about what part of the acoustic space is amenable to modification by individuals to achieve implicit vocal accommodation. However stating that this is due to 'articulatory movements' might be somewhat misleading. In my understanding, baboon grunts are fully nasal, emitted with a closed mouth. Thus, no change in oral articulators is indicated/expected here. Since 'articulators' are most commonly used to refer to oral articulators, this word might be misleading without further qualification. Also since the acoustic difference in the grunts between gangs seems relatively static, 'movement' of supra-laryngeal articulators to create these grunts within-individuals is not expected. Thus it seems likely that gang-specific grunts are produced by changes to pharyngeal or nasal space, that is relatively non-variant within-individuals, once it has been adopted.

I would therefore suggest the following: 1) keep in that acoustic variation is found in the top quartile of the acoustic space, and remain agnostic about how this is achieved; or 2) (my preferred option) state more clearly what anatomical change might account for this acoustic shift.

Review form: Reviewer 2

Recommendation

Accept with minor revision (please list in comments)

Scientific importance: Is the manuscript an original and important contribution to its field?

Excellent

General interest: Is the paper of sufficient general interest?

Good

Quality of the paper: Is the overall quality of the paper suitable?

Excellent

Is the length of the paper justified?

Yes

Should the paper be seen by a specialist statistical reviewer?

No

Do you have any concerns about statistical analyses in this paper? If so, please specify them explicitly in your report.

No

It is a condition of publication that authors make their supporting data, code and materials available - either as supplementary material or hosted in an external repository. Please rate, if applicable, the supporting data on the following criteria.

Is it accessible?

Yes

Is it clear?

Yes

Is it adequate?

Yes

Do you have any ethical concerns with this paper?

No

Comments to the Author

The authors did a great job incorporating the ideas and suggestions made in the previous revision. I only have one minor comment: The authors still sometimes use the terms vocal similarity and vocal dissimilarity (e.g. line 256 and line 261). It is not fully clear to me if the two terms are used as synonyms or to address two different things. It might be worth to either give a brief explanation for when which term is used, or to just use either term, to be more consistent. Otherwise, I would like to commend them for this interesting and important study, which provides further information about the vocal flexibility that can be found in nonhuman primates.

Decision letter (RSPB-2020-2531.R0)

21-Nov-2020

Dear Professor Fischer

I am pleased to inform you that your manuscript RSPB-2020-2531 entitled "Vocal convergence in a multi-level primate society: insights into the evolution of vocal learning" has been accepted for publication in Proceedings B. The reviewers, AE and I appreciate the efforts you put in to your revision. Each reviewer made a suggestion that I think is worth considering to improve the clarity of your manuscript, so I ask you to address them as you see fit prior to publication. Because the schedule for publication is very tight, it is a condition of publication that you submit the revised version of your manuscript within 7 days. If you do not think you will be able to meet this date please let us know.

[http://datadryad.org/submit?journalID=RSPB&manu=\(Document not available\)](http://datadryad.org/submit?journalID=RSPB&manu=(Document not available)) which will

take you to your unique entry in the Dryad repository. If you have already submitted your data to dryad you can make any necessary revisions to your dataset by following the above link. Please see <https://royalsociety.org/journals/ethics-policies/data-sharing-mining/> for more details.

Sincerely,
Dr Sarah Brosnan
Editor, Proceedings B
mailto: proceedingsb@royalsociety.org

Associate Editor
Board Member
Comments to Author:

The authors have done an excellent job revising the paper and in my view this paper will make a fine contribution to the literature on vocal learning in animals and language evolution. Both reviewers have made some final helpful suggestions to improve clarity which will increase the accessibility of the paper.

Reviewer(s)' Comments to Author:

Referee: 1

Comments to the Author(s).

I appreciated the changes the authors have made. I find the revisions clear, appropriate and well done.

I would suggest only one small change. It is very helpful to have the additional information about what part of the acoustic space is amenable to modification by individuals to achieve implicit vocal accommodation. However stating that this is due to 'articulatory movements' might be somewhat misleading. In my understanding, baboon grunts are fully nasal, emitted with a closed mouth. Thus, no change in oral articulators is indicated/expected here. Since 'articulators' are most commonly used to refer to oral articulators, this word might be misleading without further qualification. Also since the acoustic difference in the grunts between gangs seems relatively static, 'movement' of supra-laryngeal articulators to create these grunts within-individuals is not expected. Thus it seems likely that gang-specific grunts are produced by changes to pharyngeal or nasal space, that is relatively non-variant within-individuals, once it has been adopted.

I would therefore suggest the following: 1) keep in that acoustic variation is found in the top quartile of the acoustic space, and remain agnostic about how this is achieved; or 2) (my preferred option) state more clearly what anatomical change might account for this acoustic shift.

Referee: 2

Comments to the Author(s).

The authors did a great job incorporating the ideas and suggestions made in the previous revision. I only have one minor comment: The authors still sometimes use the terms vocal similarity and vocal dissimilarity (e.g. line 256 and line 261). It is not fully clear to me if the two terms are used as synonyms or to address two different things. It might be worth to either give a brief explanation for when which term is used, or to just use either term, to be more consistent.

Otherwise, I would like to commend them for this interesting and important study, which provides further information about the vocal flexibility that can be found in nonhuman primates.

Author's Response to Decision Letter for (RSPB-2020-2531.R0)

See Appendix B.

Decision letter (RSPB-2020-2531.R1)

25-Nov-2020

Dear Professor Fischer

I am pleased to inform you that your manuscript entitled "Vocal convergence in a multi-level primate society: insights into the evolution of vocal learning" has been accepted for publication in Proceedings B.

Your article has been estimated as being 8 pages long. Our Production Office will be able to confirm the exact length at proof stage.

Open Access

Paper charges

You are allowed to post any version of your manuscript on a personal website, repository or preprint server. However, the work remains under media embargo and you should not discuss it

with the press until the date of publication. Please visit <https://royalsociety.org/journals/ethics-policies/media-embargo> for more information.

Sincerely,
Editor, Proceedings B
<mailto:proceedingsb@royalsociety.org>

Appendix A

Dear Dr. Brosnan,

Very many thanks for giving us the opportunity to revise our manuscript RSPB-2020-1666 entitled "Vocal convergence in a multi-level primate society: insights into the evolution of vocal learning". We were delighted to hear that you, the AE, and the reviewers were generally enthusiastic about the goals of this study.

We substantially revised the manuscript, in light of the comments, questions, and suggestions by the AE and the reviewers. Below, we detail point by point how we responded (in blue). Please note that line numbers refer to the manuscript version with track changes. We hope that we were able to address all issues in a satisfactory manner. Once again, thanks for the insightful comments. We think that the manuscript has benefitted from the additional information and clarifications and hope that the editors and the reviewers share this view.

Yours sincerely, on behalf of all authors,

Julia Fischer

You wrote:

In particular, while any study of this sort will have both a small sample and a modest effect, there needs to be more discussion of how this impacts interpretation of the data.

See response to AE

Associate Editor

Comments to Author:

The paper presents findings suggesting that social learning of vocalisations takes place in primate societies. They report similarity of vocalisations that can't be attributed to genetics. This is interesting and has important implications for the evolution of language. However, the effect is small and (although bigger than many other studies) the sample size is still limited. In my view the authors need to comment on this and provide justification. R2 is also concerned about pseudoreplication. My reading is that the DFA takes individual into account so this shouldn't be a major concern, but the authors need to clarify this.

Thanks to the AE for this positive assessment. If we are not mistaken, we are among the first to explicitly report effect sizes. That the effects are small is part of the message: it is necessary not only to consider whether there is "statistically significant" variation between groups, but also to consider the extent of the variation. To drive home this point, we added the following sentence: "Thus, the small effect sizes reported here are an important aspect of the results." (line 327)

We also agree that the sample size is not large, but it was limited by the size of the study population (i.e. there were no more habituated males from which we were able to obtain a decent number of sound recordings), and we are unable to follow further groups without compromising the quality of the data that we are getting for each subject. We now write: "Ideally, one would like to include further groups with additional subjects to assess whether

the observed pattern hold across multiple groups, but this was beyond our capacities.” (line 153 ff)

Finally, as we explain in more detail below, non-independence should not be an issue, as only the average F-value per dyad is used in the analysis. Thus, the problem of pseudoreplication does not exist in the statistical analysis of the link between social level, genetic relatedness, and acoustic similarity. In the analysis of the assignment of calls to individuals (not a core part of this study), we additionally ran the permuted DFA to control for uneven sample sizes and uneven representation of males in the social levels. The results had been stated in the original manuscript already.

Reviewer(s)' Comments to Author:

Referee: 1

Comments to the Author(s)

This is a beautiful piece of work examining group differences in the acoustic properties of grunts in a population of guinea baboons in Senegal. The authors found that acoustic and genetic similarity did not correlate. The authors provide a clear, interesting discussion drawing from neurobiology to propose potential mechanisms to account for the findings, as well as a helpful discussion contrasting implicit and explicit vocal learning patterns.

Not addressed, but of interest, given that some implicit vocal convergence of individuals within but not between parties seems to occur, is what acoustic features seem to be more amenable to this type of plasticity than others? This information is presumably evident in the acoustic analysis conducted. It would be a useful addition to present this.

We thank the reviewer for the praise and this valuable suggestion. The analysis we conducted in response unearthed some interesting information: the “group signatures” of the grunts are most likely due to variation in articulatory movements, reflected by variation in the distribution and modulation of the amplitude in the frequency spectrum in the top quartile. We have added a Table in the Electronic Supplementary Material that lists the acoustic parameters suited to distinguish between gangs and parties, and added a general remark in the text. We now write: “Grunts varied with social level (party/gang) mostly in parameters that are related to articulatory movements (Tab. S6). The fundamental frequency or duration did not vary systematically between social levels.” (line 263 ff).

Also not addressed, but would be useful for future research in this area, is even a short discussion regarding the purported function of this behavior – what do male Guinea baboons gain from a mechanism of vocal convergence with those they engage in more social interactions? In lines 281-282, the authors suggest this may simply be a “by-product of the sensory experience and does not necessarily be selected for”. However, it seems equally parsimonious to suggest that this could have been selected for, especially as hypotheses exist for vocal convergence in other species. Whilst there may not be space to go into this in detail here, it would be important to briefly acknowledge this possibility - unless the authors feel they have a case to discount this possibility. If the latter, it would still be useful to state the author’s line of argument here.

We thank the reviewer for this suggestion and have changed this section into an open question. We now write “An interesting open question is whether vocal convergence is simply a by-product of the sensory experience or whether it has been selected for, since it may signal ‘in-group’ membership and thus have an important social function [46].” (line 333 ff).

In terms of data access, I could not find any genetic relatedness data, but perhaps I have missed it?

The information was provided in the file “Fischer_VocConv.RData” (dataset “d”). We replaced this file with a simple data file “Fischer_VocConv_Data.csv” uploaded on OSF.

Other minor comments:

L. 113. Perhaps here it would be good to add a sentence to clarify the basis for assigning units to parties and parties to gangs. Presumably this is based on association, rate of social interactions and some social network measure? It would be good clarify this. Is this essentially categorical assignment – ie some units never associate so are not assigned to the same party, or is it more a classification by degree?

We apologize that we did not explain this more clearly before. We now write: “At the base of the society are ‘units’ comprising one adult male, a small number of females and young. A small number of units, together with bachelor males, form a ‘party’, and 2-3 parties make up a ‘gang’ (Fig. 1a). Assignments to the different levels are based on spatial proximity and affiliative interactions [19].” (line 113 ff)

L. 193. If I understand correctly, the comparison of grunt similarity within and between parties is conducted only at the gang level and not between gangs? Is there an ecological basis for this eg that baboons from different gangs do not overlap spatially nor acoustically? If there is overlap between gangs, would it not strengthen the understanding of the results to compare between parties of different gangs? By this I mean that a) if vocal differences between parties arise due to frequent social interactions, then this result should be strengthened by comparing across parties of different gangs. B) if vocal differences between parties drop when including parties from different gangs, this is also interesting.

We acknowledge that it would be technically possible to analyse the similarity of grunts from party A in gang 1 to those of party C and party D in gang 2, and then those of party B in gang 1 to C and D, and so on, but this would create a problem of multiple testing of the same hypothesis with different combinations of data. Moreover, as parties are nested within gangs, we are in fact testing the variation between parties in different gangs. We hope the reviewer can accept that we did not run the suggested analyses.

L. 254. Given that the effect size is small perhaps ‘higher’ acoustic similarities between parties, is more accurate than ‘high’.

Very true. This part of the text has been rewritten.

L. 282. Word missing in this sentence?

Thanks for spotting this mistake. This section has been revised.

L. 334. In the conclusion, you say with what your results in the Guinea baboons do not compare with in humans. I suggest it would be helpful to add here – as it would be in the abstract - what it does compare with in humans ie implicit vocal accommodation, as seen when acquiring speech accents/dialects.

Thanks for this suggestion; we now write in the Abstract: “We consider this convergence a result of sensory-motor integration and suggest this to be an implicit form of vocal learning shared with humans, in contrast to the goal-directed and intentional explicit form of vocal learning unique to human speech acquisition.” (line 37 ff)

Similarly, in the discussion, we now say: “... we propose that vocal convergence (or vocal accommodation) is an implicit form of motor learning shared between nonhuman primates and humans, while speech production constitutes an explicit form of motor learning.” (line 338 ff)

Referee: 2

Comments to the Author(s)

The authors studied whether vocal similarity in Guinea baboon males is more likely explained by social unit or by genetic relatedness. This is especially interesting as a higher vocal similarity within social unit, independent of relatedness, hints at some vocal flexibility often denied to exist in primates. What the authors found was that indeed vocal similarity was higher within units than between units, and that this seemed to be the case independent of the relatedness between individuals, even though relatedness was also higher within social unit than between.

In general, I liked research question as well as the approach of the study, but I think there are quite a few things that could be clarified both in the manuscript as well as in the analysis to help understanding it even better, see comments below.

We are happy about the overall positive assessment and tried to clarify the important points raised by the reviewer (see below).

Abstract, Line 33: Does social level here refer to the social organization? Spontaneously, as not being a baboon expert, I would have assumed it to be hierarchical levels.

To avoid confusion, we gave more details on the nested structure of the society and replaced “social level” with “party/gang membership”. We changed the text accordingly: “Guinea baboons live in a multi-level society, with units nested within parties nested within gangs. We investigated whether the acoustic structure of grunts of 27 male baboons of two gangs varied with party/gang membership and genetic relatedness.” (line 28 ff)

Line 80: common marmosets also show evidence for adjusting to social environment (see Zürcher, Willems & Burkart, 2019, Plos One)

Thanks for reminding us of this important paper. We incorporated it into the discussion, as the results reported can be well interpreted within the framework suggested here. We now write: “In a translocation experiment, common marmosets (N = 4) adjusted the structure of

their vocalizations in response to auditory input from conspecifics, even if the individuals do not interact directly [36].” (line 306 ff)

Line 115ff: The affiliative interactions, during which males produce grunts, are they usually directed towards other males, or towards female and offspring? And do females also grunt? I think this is highly important to know. If males mainly grunt towards females, the grunts would not be social interactions directed towards the males, which I think would be the ones we would expect to have an influence on the male’s vocalization.

We now give the following information in the manuscript: To maximize the number of grunts per male, we included grunts from different contexts (Electronic Supplementary Material Tab. S2). In total, we included 756 grunts in the acoustic analysis.” (line 149)

In the ESM, we provide a table with the number of calls per male and context (Tab. S2), and a description of the contexts. Given the minor variation in relation to context, and the fact that the majority of males was represented in multiple contexts, we assume that context differences do not contribute to the observed variation between social levels.

Line 135: Were all males present during the whole study? Were there no changes within the unit or party-composition by males becoming adults or dying? If it was, how was this taken into account in the analysis?

The majority of males was confirmed or assumed present throughout the study. We added a supplementary table that shows the presence and whether recordings were taken for each of the three recording periods (new Table S1 in ESM).

Line 146ff: What was the reason to perform two FFT’s? How where the results of the two analyses treated afterwards? Where they somehow merged or both included in the following analyses?

We added the explanation: “Calculating two FFTs allowed us to maximize the temporal resolution for the entire call type, and estimate the fundamental frequency at a higher frequency resolution.” (line 162 ff)

Line 156: “...all of which were uttered in affiliative contexts”. Do you have any further information of the context in which they were uttered? Were they all mixed? Again, I think the communication partner here could be quite important.

Please see above

Line 159ff: When quantifying the acoustic similarity, did you use a call average for each male, or how did you take the non-independence of the calls into account when calculating the F-values?

The F-values used in the matrix correlations are the average dissimilarity for all pairwise comparisons of all of the dyad males’ grunts. In other words, there are multiple comparisons for each dyad that are then averaged. Non-independence should not be an issue at this stage, as only one value per dyad is ultimately entered into the analysis.

Line 162: In the supplementary table, I count around 30 parameters. How do you get to 82 in this analysis? And were the parameters transformed in any way to take into account different measurement units?

We now clarify that the LMA software developed by Kurt Hammerschmidt by default outputs 82 variables. These are then submitted to the parameter selection procedure in the stepwise discriminant analysis in SPSS. We provide the description of the 31 selected variables in the supplementary table. The parameters were not transformed.

Line 188: For me, it was somewhat hard to understand on which levels you test the correlation between acoustic and genetic variation. Probably this could be re-formulated a bit to help the readers understand the process better?

We rewrote this section and expanded it slightly. Hopefully, the approach is now clear. We write: “To test whether calls a gang were more similar to each other than between gangs, we applied a categorical mantel test, using ‘same gang membership’ (Yes/No) as the categorical predictor variable, and W or F (transformed as $\ln(1+F)$) as the continuous variable. The analysis of the effect of gang membership was based on 351 dyads. To study the effect of party membership, we also used a categorical mantel test, but only considered pairs of males that lived in the same gang (e.g., ADM and ASN who are both members of the Mare gang, or BEN and WLD who are both members of the Simenti gang; total number of dyads within both gangs $N = 169$). We used a restricted permutation approach where males were permuted between parties within gangs.” (line 207 ff)

Line 194: If I understand this correctly, you only compare the more closely related individuals with each other, by only including the dyads within the same gang. Wouldn't comparing between less closely related individuals actually be more interesting and more meaningful, as there would be more variation in relatedness within which differences could be found? I would expect that, if you find a correlation between relatedness and vocal similarity, it would be between the less related parties of different gangs.

We are sorry that this did not become fully clear. In the analysis presented in the first version, we compared all dyads, irrespective of relatedness. We now added a further analysis in which we selected highly related and unrelated subjects only, and then measured whether these differed with regard to acoustic similarity. There were no significant differences ($P=0.933$). The previous correlation plot was replaced (new Fig. 3).

Methods in general:

What about the units? As they are the structure where animals interact the most, it would be interesting to know why you did not include them into your analyses?

We now clarified that there is only one male in each unit. (line 113)

Line 223: When checking the “baboons 27males pdfa.txt” file, it shows a different value for percent correctly cross classified (approx.11.21%). Probably worth checking again.

These are two different procedures. The one reported in the text is the one done using SPSS, with the full data set. The pDFA selects only a subset for each permutation, and yields a significantly lower average classification (as noted: 11.21%). We added this information in the text. (line 237 ff)

Line 230: This is the first time acoustic dissimilarity comes up. Could you quickly explain what you mean by it and how you calculated it?

Thanks for pointing out this omission. We now provide the explanation in the methods section. We write: "To quantify the acoustic variation between males, we used a discriminant function analysis with which we obtained a pairwise F-value (dissimilarity score) for the acoustic distance for each dyad." (line 173)

Line 231: there is something missing after SD=, as well as the closing bracket

Sorry, something had gone wrong here. We have corrected this and now provide the mean estimates for both social levels. We write: "When we compared the mean acoustic dissimilarity of males that resided in the same party (mean logF = 0.29) to those that were part of a different gang (mean logF = 0.33), the effect size was small ($d = 0.24$, $CI_{lower} -0.02$, $CI_{upper} 0.50$)." (line 260 ff)

Line 250: Your results are indeed somewhat confusing. Could you elaborate on some ideas here on why this could be the case? I think it would really be worth it to help the reader along, as this is your main result of the study, but it is really not easy to grasp.

We tried to better unpack the problem. We now write: "It may seem puzzling at first that genetic relatedness did not account for the higher vocal similarity in Guinea baboons despite the fact that genetic relatedness and acoustic similarity were both higher within parties and gangs than between. This can be explained by a combination of the fact that not all dyads within a social level are indeed more highly related than across these social levels, and that acoustic similarity appears mainly to be driven by social interaction, which is not restricted to highly related dyads. To a certain degree, relatedness and acoustic similarity vary independently of one another." (line 284 ff)

Line 267: See also Gultekin & Hage 2017 on how auditory feedback can be important for proper vocal development.

We now cite Gultekin & Hage 2017 and Gultekin & Hage 2018, as well as Takahashi et al. 2017 and state: "Beyond the immediate effects of auditory experience, there is also evidence that feedback from parents affects the trajectory of vocal development in marmosets [38–40]." (line 309 ff)

Line 257 – 316: Even though highly interesting, I find this part in the discussion a bit too long, especially as it is not really linked to the study or the results themselves. Probably this could be condensed a bit in favour of providing potential explanations for the results?

Point taken. We have shortened the part of the discussion of sensory-motor integration in the visual domain, and restrict ourselves to the auditory-motor domain here. The space is used to unpack the results some more.

Appendix B

Dear Dr. Brosnan,

We are delighted that you now found the paper acceptable for publication. We have responded to the remaining queries in the following way:

Referee: 1

Comments to the Author(s).

I appreciated the changes the authors have made. I find the revisions clear, appropriate and well done. ...

I would therefore suggest the following: 1) keep in that acoustic variation is found in the top quartile of the acoustic space, and remain agnostic about how this is achieved; or 2) (my preferred option) state more clearly what anatomical change might account for this acoustic shift.

We hope that you find it acceptable that we opted for version 1 and remain agnostic. We now simply write: "Grunts varied with social level (party/gang) mostly in parameters that are related to the filter function of the vocal tract." (line 266 f).

Referee: 2

Comments to the Author(s).

The authors did a great job incorporating the ideas and suggestions made in the previous revision. I only have one minor comment: The authors still sometimes use the terms vocal similarity and vocal dissimilarity (e.g. line 256 and line 261). It is not fully clear to me if the two terms are used as synonyms or to address two different things. It might be worth to either give a brief explanation for when which term is used, or to just use either term, to be more consistent.

Otherwise, I would like to commend them for this interesting and important study, which provides further information about the vocal flexibility that can be found in nonhuman primates.

We changed the text as follows (line 173ff, critical parts bold):

"To quantify the acoustic distance between males, we used the average pairwise F-value from the discriminant function analysis as a dissimilarity score for each dyad. The dissimilarity score provides an assessment of the similarity of calls, **with higher values indicating greater dissimilarity and lower values greater similarity. In the following, we will simply refer to the similarity of calls.** The average pairwise F-value has been used in different studies examining relationships between acoustic structure and genetic or geographic distance [22,23].

We hope that no further amendments are necessary and would like to thank you and the reviewers for a highly constructive review process.

Best wishes

Julia